# Molecular and Biochemical Techniques for Deciphering p53-MDM2 Regulatory Mechanisms

**DOI:** 10.3390/biom11010036

**Published:** 2020-12-30

**Authors:** Konstantinos Karakostis, Ignacio López, Ana M. Peña-Balderas, Robin Fåhareus, Vanesa Olivares-Illana

**Affiliations:** 1Inserm UMRS1131, Institut de Génétique Moléculaire, Université Paris 7, Hôpital St. Louis, F-75010 Paris, France; chem898@yahoo.gr (K.K.); robin.fahraeus@inserm.fr (R.F.); 2Biochemistry-Molecular Biology, Faculty of Science, Universidad de la República, Iguá 4225, Montevideo 11400, Uruguay; lopez@fcien.edu.uy; 3Laboratorio de Interacciones Biomoleculares y Cáncer, Instituto de Física Universidad Autónoma de San Luis Potosí, Manuel Nava 6, Zona Universitaria, San Luis Potosí 78290, Mexico; annabalderas25@gmail.com; 4Regional Centre for Applied Molecular Oncology (RECAMO), Masaryk Memorial Cancer Institute, Zluty Kopec 7, 65653 Brno, Czech Republic; 5Department of Medical Biosciences, Building 6M, Umeå University, 90185 Umeå, Sweden; 6International Center for Cancer Vaccine Science (ICCVS), University of Gdańsk, Science, ul. Wita Stwosza 63, 80-308 Gdańsk, Poland

**Keywords:** protein-protein interactions, protein-RNA interactions, p53 mRNA, MDM2, p53, MDMX, ATM, post-translational modification, DNA damage response

## Abstract

The p53 and Mouse double minute 2 (MDM2) proteins are hubs in extensive networks of interactions with multiple partners and functions. Intrinsically disordered regions help to adopt function-specific structural conformations in response to ligand binding and post-translational modifications. Different techniques have been used to dissect interactions of the p53-MDM2 pathway, in vitro, in vivo, and in situ each having its own advantages and disadvantages. This review uses the p53-MDM2 to show how different techniques can be employed, illustrating how a combination of in vitro and in vivo techniques is highly recommended to study the spatio-temporal location and dynamics of interactions, and to address their regulation mechanisms and functions. By using well-established techniques in combination with more recent advances, it is possible to rapidly decipher complex mechanisms, such as the p53 regulatory pathway, and to demonstrate how protein and nucleotide ligands in combination with post-translational modifications, result in inter-allosteric and intra-allosteric interactions that govern the activity of the protein complexes and their specific roles in oncogenesis. This promotes elegant therapeutic strategies that exploit protein dynamics to target specific interactions.

## 1. Introduction

Protein-protein interactions (PPIs) are involved in all aspects of cellular functions. The identification and characterisation of PPIs are essential for understanding the molecular mechanisms that regulate biological systems and for guiding drug design programs [1,2,3]. PPIs can modify the kinetic properties of enzymes, form new binding sites, control the localisation, and change the specificity, among others [4]. The spatial arrangement of protein complexes is determined by the composition of the amino acid of the proteins involved, their concentration, and the free energy of the complex.

Studies on PPI networks and their topologies revealed the existence of nodes of proteins able to interact with a large number of partners [5]. These proteins are known as hubs and they have particular biological properties. They tend to be evolutionarily conserved to a larger extent compared to non-hubs [6]. Their ability to interact with multiple partners is often facilitated by intrinsically disordered regions (IDR), and it is regulated by post-translational modifications (PTMs) that govern different conformations and bound states [7,8]. Such conformational changes lead to allosteric structural modifications, which dynamically regulate the interactions with binding partners. Allosterically-induced interactions are vital for the regulation of several proteins, determining their function and orchestrating the physiological effect of the cognate signalling pathway [9]. Since hubs play central roles in signalling networks, they constitute exciting targets for drug development applications.

The p53-MDM2 pathway is a dynamic and well-characterised model used to study both protein-nucleic acid and PPIs since both Mouse double minute 2 (MDM2) and p53 are hubs in a vast network of interactions involved in several cellular pathways [10,11,12]. The p53 tumour suppressor is a crucial regulator of cellular homeostasis and it is tightly linked to cancer development. p53 function is lost in more than 50% of all types of human cancers and represents a main target of genetic diagnostics and therapeutic interventions. As a result, the p53 pathway regulation constitutes an ideal study-model for improving the current state of methodologies aiming to decipher the underlying mechanisms. Under normal conditions, p53 activity is low due to its interaction with MDM2, which exhibits an E3 ubiquitin ligase activity that targets p53 for degradation via the 26S proteasome, and with its homolog Mouse double minute 4 (MDMX) that blocks the transcriptional activity of p53 [13]. Following genotoxic stress, p53 levels increase to allow the cells to repair the damage before entering replication, or to trigger irreversible senescence or apoptosis, when the damage is too severe. Activation of p53 requires MDM2 to switch from binding the p53 protein to *p53* mRNA and become a positive regulator of the p53 tumour suppressor protein. This involves the ATM kinase-dependent phosphorylation of MDM2 on Ser 395. This PTM induces a conformational change on MDM2, which results in allosteric changes allowing the formation of a *p53* mRNA-binding site that stimulates p53 synthesis [14,15]. Together with ribosomal proteins, such as RPL5 and RPL11, the complex (MDM2-*p53* mRNA-RPs) is transported to the cytoplasm where the p53-polysome is formed. ATM also phosphorylates MDMX at Ser 403, which promotes its RNA chaperone activity toward the *p53* mRNA to create an mRNA structure suitable for the MDM2-*p53* mRNA interaction [16]. Given this complexity, it becomes clear that the mechanistic description of the p53 regulation requires the application of multi-faceted techniques and methodologies. In addition, the study of p53 regulation needs be approached from several levels, including (a) in vivo, in vitro, and in situ techniques, adequately addressing the interactions and the expression levels, and (b) the effect of different conditions that alter those interactions and the expression of the partners involved. Diverse studies have employed proteomic techniques in an attempt to broadly identify binding partners and to unravel the mechanisms control, which are regulated by the p53-MDM2 pathway [17,18,19,20]. However, in this review, we focus on well-studied mechanisms regulating the p53-MDM2 pathway, and on the experimental methodology that has been building up for past years for revealing the mechanism whereby p53 is activated during the DNA Damage Response (DDR), rather than list binding partners, which can be found in the literature and in databases, such as the BioGRID.

## 2. PPI Classic Techniques

The broadly used co-immunoprecipitation (CoIP) assay and the enzymatic immunoassay (enzyme immunoassay EIA or enzyme-linked immunosorbent assay ELISA) may be performed in vitro, or ex vivo using cell extracts [21,22]. They both have significant advantages such as a low-cost, the ease of use and, most importantly, the possibility to study endogenously-expressed protein complexes while avoiding negative side-effects of overexpression or addition of tags [21,22,23]. CoIP can be coupled to Western blotting to detect a specific interactor, to mass spectrometry in high-throughput settings, and linked to qRT-PCR to detect protein-nucleic acids interactions in techniques such as CoIP-RNA and chromatin immunoprecipitation. However, low-affinity interactions studied with CoIP might require using customized conditions by modifying the amount of the interacting proteins in the lysate in order to allow the detection. This may lead to artificial and inconclusive results with respect to the in vivo physiology. Since cells are disrupted, CoIP does not reveal the in-situ localisation of the interactions and has a significant risk of detecting false-positives due to artificial or biologically non-relevant associations [21,23]. CoIP has enormously contributed to the early characterization of the p53 pathway. Furthermore, p53 was first detected when it was co-immunoprecipitated along with SV40 large T and small t antigens in SV40-infected and transformed cells [24,25]. Furthermore, both MDM2 and MDMX were described to bind p53 using CoIP experiments [26,27,28] (Table 1). CoIP was also used to detect that the amount of MDM2 bound to p53 remained constant after DNA damage, though levels of both p53 and MDM2 increased upon stress induction. This observation was later linked to the DNA damage-dependent phosphorylation of p53 on serine 15 that promotes a conformational change on p53 that impairs MDM2′s ability to bind and inhibit its activity [29]. Currently, CoIP serves as an essential tool both for confirming and detecting new PPIs.

Although CoIP is used to estimate the relative amount of proteins in complexes, more accurate quantifications are performed by EIA/ELISA since the signal is proportional to the quantity of the antigen in the sample [21]. Due to the characteristics of the antibody-antigen interaction, EIA/ELISA is a highly specific and sensitive method. These properties, along with the brief time required to perform the assay, explain its wide use in diagnostic tests of diverse diseases as well as in biochemical research [21]. EIA/ELISA has been widely used to characterise the p53-MDM2 interaction. Using synthetic and p53-derived peptides from phage display libraries, EIA/ELISA was applied to confirm the role of residues F19, W23, and L26 as critical contact points on p53 when interacting with MDM2 [30] (Table 1), as previously suggested by the crystal structure of the p53-MDM2 interface [27]. In addition, EIA/ELISA was applied to reveal the effect of the NO-induced reversible oxidation of a conserved Cys77 residue on MDM2′s capacity to interact with p53. Although not involved in a direct contact with p53, oxidation of Cys77 provides an explanation for the oxidative stress-dependent activation of p53, exemplifying the impact of altering the conformation of p53 and MDM2 [31] (Table 2). EIA/ELISA allowed us to demonstrate that, although the MDM2 S395D phosphomimetic mutant is able to bind the p53 protein, the presence of the *p53* mRNA blocks the PPI, bringing forth evidence for the positive role of phosphorylated MDM2 toward p53 activity during DNA damage [14] (Table 1). Despite the utility of EIA/ELISA, one important caveat is the impossibility to directly detect the sub-cellular location where the interactions occur.

The shared localisation (co-localisation) of partners within the cell at a given time is necessary but not sufficient for PPIs to occur. As such, immunofluorescence results merely describing the localisation may potentially lead to erroneous interpretations about putative PPI. Light microscopy is very useful to dissect the sub-cellular distribution of proteins and to evaluate co-localisation of different targets, but it lacks the resolution to identify the juxtaposition of two molecules based on their distribution in fluorescence images [46,47]. Nevertheless, the use of fluorescence microscopy has greatly contributed to the dissection of the p53 pathway. It has granted the visual means to identify the cluster of several nuclear localisation signals contained on the C-terminal domain of p53 [48,49], and to define that the MDM2-dependent ubiquitination and proteasomal degradation of p53 can take place both in the nucleus and cytoplasm [32,33]. On the other hand, tumour suppressor Alternative Reading Frame p14 (ARF)-mediated stabilization of p53 in response to DNA damage requires the sequestration of MDM2 into the nucleolus. This process is controlled by the nucleolus retention sequence of ARF that was shown by the localisation of ARF mutants using fluorescence microscopy [34,35] (Table 1).

The regulation of p53-MDM2 interaction was also studied in vivo using techniques that depend on a readable output upon fusion or close encounter of two different probes. Based on the nature of the association, three main categories are defined: (a) protein fragment complementation assays (PCA), (b) assays based on a resonance energy transfer (RET), and (c) the two-hybrids assay [22].

Several variations of PCAs have been developed in the past 30 years, including the split ubiquitin system or ubiquitin-based split-protein sensor [50], the split-luciferase (and other enzymes including galactosidase and beta-lactamase) complementation assay [22,51], and the bimolecular fluorescence complementation assay (BiFC) [52]. The latter is based on the use of two non-fluorescent fragments of a fluorophore (such as GFP) linked to the proteins of interest that fluoresce once in close proximity [52]. Modulation of the p53-MDM2 interaction has also been studied by BiFC. This is the case of ceramide that binds to the secondary contact region of p53-MDM2 located at the box V motif and disrupts the p53-MDM2 complex, leading to p53 accumulation and activation [42]. Ceramide constitutes an example among few other metabolites that directly bind to p53 to regulate its stability. In addition, the mechanism was shown to be triggered by serum or folate deprivation, adding to the link between metabolic and nutrient stress to the p53 pathway. BiFC has also contributed to characterise the capacity of MDM2′s RING domain to bind RNA and the effect this interaction has on MDM2 stabilisation [38] (Table 1). The presence of *Xiap* IRES decreased MDM2 homodimerization and, subsequently, MDM2 self-ubiquitination and degradation, resulting in inhibition of p53 and induction of cell growth and survival [38]. As with the interaction with the *p53* mRNA, this study highlights the important role of establishing MDM2′s RING domain-RNA complexes on altering MDM2 conformation and activity, and on the down-stream cellular output. Due to the irreversible complementation of the fluorescent fragments, BiFC does not offer a dynamic analysis of the interactions. However, this is a positive trait when BiFC is coupled to flow cytometry since it stabilizes weak associations for proper detection and quantification [22,53].

More precise relative quantification assessments can be carried out using RET-based assays. RET occurs when a portion of the energy of an excited donor is non-radiatively transferred to a nearby acceptor molecule. This requires an overlap of the emission and absorption spectrums of the donor and the acceptor, and an aligned relative orientation at a permissive distance of 10 to 100 Å, depending on the assay [54]. In Förster Resonance Energy Transfer (FRET) both donor and acceptor, are fluorescent proteins with different excitation spectrums. The use of a monochromatic light at the appropriate wavelength is required to excite the donor [54,55]. FRET has been used to identify different types of MDM2 and MDMX inhibitors. This is the case of the dual small molecule RO-5963 that binds to both MDM2 and MDMX and induces the formation of a dimeric complex that is kept together by the inhibitor and that is not able to interact with p53. Thus, formation of this inactive heterodimer results in p53 stabilization and induction of the cell cycle arrest and apoptosis in different cancer cells [44]. Other small activators of p53 with different molecular mechanisms were found in a FRET-based E3 ligase activity assay. MMRi64, for example, disrupts the MDMX-MDM2 RING-RING interaction, which leads to inhibition of MDMX-induced MDM2 E3 ligase activity toward p53, resulting in activation of the apoptotic arm of the p53 pathway in leukaemia/lymphoma cells [43,44] (Table 2). Identification and characterization of p53 partners and regulators have also been accomplished by the use of FRET. For example, detection of FRET by confocal microscopy was used to confirm the interaction between ribosomal protein S3 and p53 or MDM2 [39], adding more evidence toward the importance of ribosomal proteins in controlling the p53 pathway. Coupling FRET with different microscopy approaches offers the possibility to detect PPIs in cells or tissues. However, the use of external light as a source of energy is a limitation since it leads to photobleaching of the acceptor, thus, lowering the detection threshold, or it results in autofluorescence of the sample, increasing the background noise [55]. These issues were partially circumvented by the introduction of BRET.

In Bioluminiscence Resonance Energy Transfer (BRET), the transferred energy is bioluminescence and the external source of energy is provided by the oxidation of the donor’s substrate (cœlenterazine). The targets are either fused to *Renilla* luciferase (donor) or a fluorescent protein such as the YFP (acceptor) and the interaction is measured by an increase in the acceptor’s emission [55,56]. BRET was used to show that the shorter p53/47 isoform establishes oligomers more easily than the p53 full-length protein. Since this isoform is specifically induced during endoplasmic reticulum (ER) stress and has a different set of target genes compared to the p53 full-length, the preferential formation of p53/47 oligomers points toward an additional level of regulation in this scenario. This may help explain the distinctive cellular outputs promoted by these two isoforms and highlights the role of the trans-activation domain I on p53′s conformation and activity (Table 1) [57]. Despite recent efforts to produce more sensitive devices and probes with stronger emission capacities, the assessment of PPIs in situ by BRET remains cumbersome due to the weaker signal obtained, as compared to FRET probes. In addition, the mandatory tagging and overexpression of proteins may lead to biologically non-relevant results. Nonetheless, the low background in the BRET signal and its simple implementation attest for the convenience to use it for high-throughput screening of drugs in living cells [55] and the p53-MDM2 interaction has served as a model system to explore such a capacity using the well-known MDM2-inhibitor Nutlin-3a [45] (Table 2).

The yeast-two hybrid (Y2H) method relies on the formation of a chimeric GAL4 transcription factor from *Saccharomyces cerevisiae.* GAL4 consists of two separable domains: an N-terminal domain that binds to specific DNA sequences and a C-terminal one that activates transcription from the *GAL1* upstream activating sequence (UAS_G_) [58]. Interaction of two hybrid proteins, each one containing one of the GAL4 domains, results in transcription of genes downstream of the UAS_G_ [58]. This assay has contributed toward identifying vital p53 regulators, such as MDMX [36]. Moreover, ribosomal protein S7 was shown to bind MDM2 and abrogate MDM2-mediated p53 ubiquitination, which results in stabilization and activation of p53 and apoptosis induction in cancer cells [37]. The association between S7 and MDM2 occurs when cells face ribosomal stress, which is another cell condition that impinges on MDM2-p53 interaction and adds more evidence to the intimate link between p53 and regulation of the translation machinery. Y2H has also unveiled the role of the p53 oligomerisation domain in vivo [59] and the p53 DNA-binding domain in stabilising the interaction with MDM2 [40]. Moreover, Y2H has allowed testing in vivo the capacity of different molecules to completely inhibit the p53-MDM2 interaction [40] (Table 2). An alternative version of the Y2H relying on the DNA-binding domain of GAL4 was adapted in mammalian cells. The induction of transcription from the downstream *Firefly* luciferase reporter gene is driven by the trans-activation domain of nuclear factor kappa B (NFκB). This approach allowed the discovery of the p53-MDM2 inhibitor SL-01 that causes growth arrest in tumour cells [41] (Table 2). The same rationale is applied in the microscopy-assisted fluorescent two-hybrid (F2H) assay. However, the readout is the co-localization of GFP-pray and RFP-bait on an interaction platform in the nucleus of mammalian cells. This platform is formed by a chromosomally integrated lac operator array that is bound by the LacI domain, which is also a constituent of the three-protein bait complex [60,61]. F2H allows easy assessment of static endpoint and dynamic PPIs in living cells. Using F2H, several cell-penetrating compounds were shown to potently inhibit p53-MDM2 interaction without affecting binding of p53 to MDMX [62], which is in agreement with the effect of previously-reported antagonists that are several magnitudes less potent in disrupting the p53-MDMX interaction when compared to the p53-MDM2 counterpart [63]. This is explained by the structural and dynamic differences between the two proteins and the specificities of the binding of p53 to them [64,65]. Although two-hybrid technologies are limited by the obligatory (natural or forced) nuclear distribution of the interacting partners, it is still one of the most prominent assays to detect and confirm PPIs.

## 3. Novel Techniques

### 3.1. Fluorescence Cross-Correlation Spectroscopy (FCS)

Apart from the classic techniques, new efforts have been made to study PPIs in real-time on single living cells. FCS is a quantitative and extremely sensitive technique for determining the movements and interactions of biomolecules, at the level of single living cells. The analysis of FCS data examines the minute fluorescence intensity fluctuations induced by a low number of labelled molecules that diffuse, which is caused by spontaneous deviations from the mean at thermal equilibrium in a confocal setup [66,67]. In 2018, for example, Du et al. used FCS with a microfluidic chip to monitor the p53-MDM2 interaction [68]. By using a series of p53-EGFP mutants and MDM2-Cherry constructs, the authors dissected the oligomerisation state of p53 as well as the MDM2-p53 binding by measuring the fluorescence signal fluctuation. They found that MDM2 did not bind the p53 monomer, but, instead, it stably interacted with p53 dimers, and more efficiently with p53 tetramers. The authors also confirmed the blocking capacity of Nutlin-3a and MI773 toward the p53-MDM2 interaction in living cells. On the contrary, the compound Reactivating p53 and Inducing Tumor Apoptosis (RITA) was unable to interrupt this interaction. This technique also allows studying protein-RNA interactions, with the employment of a molecular beacon hybridising with the targeted RNA [69] (Figure 1a) (Table 3).

### 3.2. Proximity Ligation Assay (PLA) and the Proximity Ligation ELISA (PLE)

The PLA is introduced as a pioneer in situ technique of significant technical advantages with respect to sensitivity and quantitative determination of endogenous interactions [71,72,73]. It exhibits a particularly high specificity, which relies on the employed antibodies and a high sensitivity, as the detection is based on an in-situ amplification of a nucleotidic molecule by isothermal PCR. Two interacting partners, which are less than 40 nm apart, can be recognised by a set of antibodies each linked to a set of specific oligonucleotide molecules that hybridise with the respective nucleotide targets (aptamers) (Figure 1b). Such aptamers are synthesized to allow a ligation reaction between them, followed by a circular DNA polymerisation reaction, which is detectable by fluorophores. The amplification step offers a detection sensitivity of proteins at concentrations of zeptomole (10–21 mol) [74]. Some examples of PPIs detected with this technique are summarised in Table 3. The versatility of PLA is shown by the several modified PLA versions that have been presented to allow the detection of PTMs [75]. Moreover, a ‘streamlined circular PLA’ version was presented as an optimised method to study molecular interactions for clinical diagnostics, such as in human plasma [76,77]. Accumulative findings suggest that the PLA concept is an established powerful tool for studying molecular interactions at the single-cell level. It is also a suitable method for applications in molecular diagnostics and for the detection of biomarkers in patient serum or blood by using two antibodies from different species against the same protein.

The PLE is a novel quantitative technique derived from the PLA. It is based on the amplification and detection of the signal generated from interactions among three binding partners (Figure 1c) [78]. Similar to PLA, PLE combines a high antibody-based specificity, but with an increased detection sensitivity due to a pre-amplification step that enhances the signal. As such, it can be applied on endogenous targets, eliminating the need of over-expressed tagged proteins. PLE offers the pioneer prospect of detecting trimeric interactions, allowing the spatial determination of the interactions at the subcellular level, when the PLE targets are exclusively expressed in specific compartments, or when it is combined with cell compartmentalisation/fragmentation.

PLE was developed and successfully applied to detect interactions that occur on the p53-polysome and the phosphorylation of serine-15, taking place on the p53 peptide while it is being synthesised (Figure 1c) [78,79,80]. The presence of ATM at the p53-polysome leads to the phosphorylation of the nascent p53 peptide at the Ser 15, which prevents its binding to MDM2 and, therefore, activates it toward the DDR, adding more evidence to their respective roles in promoting the rate of translation of p53 and its stabilisation and activation [15,81,82]. A single synonymous mutation in the *p53* mRNA (L22L) that prevents MDM2 binding, was shown to prevent p53 stabilisation following DNA damage [78,83]. However, the localisation and the mechanism could not be addressed by conventional methods. PLE was able to show that the *p53* mRNA has an effect on the encoded protein on its own polysome, during p53 synthesis. Applying PLE on fractionated polysomes extracted from cells, using a capture goat anti-RPL5 antibody and a set of primary antibodies for p53 and MDM2, showed that the interactions take place on the p53-polysome. Those results have significantly contributed to the description of the model, by localising the MDM2-nascent p53 interaction at the polysome (trimeric interaction), and accredited the description of the mechanism whereby MDM2 positively regulates p53 during genotoxic stress. This method paves the way for quantitative analysis of PPIs of low abundance, involving up to three binding partners and pioneers the detection of nascent protein interactions and modifications involved in cell signalling. The described PLE approach can be used to identify translation factors on any captured polysome and even distinguish those proteins that are present in the monosomes or the pre-initiation complexes from those within the elongating 80S ribosome.

Variations of PLE can also be applied for early detection of pathological conditions and in follow-up therapy from clinical samples, like for the detection of proteins at trace levels released from damaged tissues. Several PLE modifications have already started to be applied, including the employment of microparticles (simultaneous recognition of target proteins by three antibodies) [84]. PLE variations may couple alternative read-outs, such as qRT-PCR or next-generation sequencing (NGS), thus, offering multiplex versions to simultaneously detect multiple targeted interactions [85]. For example, a proximity ligation–based multiplexed method was employed to co-detect several proteins via unique nucleic-acid identifiers that were quantified by qRT-PCR, facilitating the scalability of the method, which is a requirement for high-throughput techniques and for the validation of new biomarker candidates [86,87].

## 4. Conclusions

Dynamic allosteric regulation allows the cell to control the interactions of a diverse number of pathways in response to cellular stress conditions or exogenous inducers. As illustrated here, PTMs are vital tools for promoting allosteric modifications that open for new interfaces which, in turn, lead to further allosteric changes, altering secondary interfaces and building up complexes for highly specific regulatory signalling system. These phenomena are widespread within the p53 pathway and are key controllers of its activity but are common among IDR-containing proteins. p53 is itself a fascinating molecule with dozens of PTM sites and hundreds of ligands that together govern its tumour suppressor and transcription factor activities. p53 and MDM2 have evolved together and these PTMs and ligands not only have inter-allosteric effects but they also impose intra-allosteric changes. This makes the p53-MDM2 a particularly challenging model system that requires the combination of several in cell, in vitro, and in-situ techniques to address its multifaceted nature (Figure 2). The methods described here constitute technical cornerstones, which drove the current knowledge on p53 and MDM2 regulation and will help to guide the studies of other IDR-carrying protein complexes.

## 5. Perspectives

The vast majority of cellular tasks are carried out by multi-protein complexes and targeting specific protein-protein interactions forms an important part in the search for novel therapeutics. Interfaces are sometimes flat and poor drug targets but, by understanding allosteric interactions, it is possible to target specific interactions with small molecules without aiming for the interface. The automatization and rapid combination of molecular and biochemical techniques remain a challenging task. It constitutes a priority for translational applications requiring evidence-based experimental findings to support genomic associations and clinical studies, given diagnostic and therapeutic applications. To this context, determining how one interaction affects the next to induce specific conformations, provides with the necessary insights for precisely targeting structural epitopes of PPIs with small chemical compounds. Such approaches targeting transitory interfaces may lead to efficient and less invasive therapies. From the mechanistic viewpoint, aiming to investigate regulatory networks, the described combined methodologies, may address qualitatively and quantitively the involved interactions as well as the tempo-spatial patterns and the responses to exogenous factors. The p53-MDM2 regulation mechanism provides with a fine model, which, over the past decades, has triggered the development and refinement of several techniques described here, which are continually leading to improved approaches and technologies, that favour the sophistication of the available molecular toolsets.

## Figures and Tables

**Figure 1 biomolecules-11-00036-f001:**
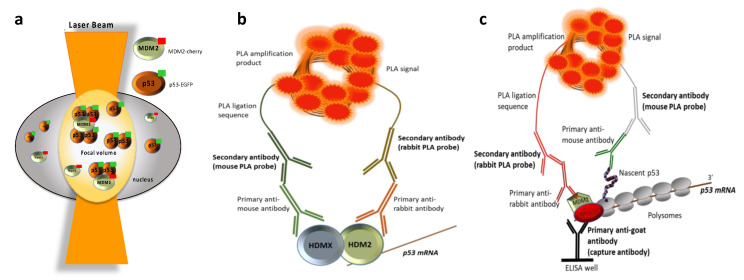
Illustrations of novel techniques used to detect interactions in the p53 pathway. (**a**) Fluorescence cross-correlation spectroscopy (FCS) used to study the p53-MDM2 interaction in living cells. FCS uses proteins that are labelled independently with different fluorescent probes. Analysis of the FCS data examines minute fluctuations in fluorescence intensity induced by a low number of diffusing labelled molecules, caused by spontaneous deviations from the mean in thermal equilibrium in a confocal configuration [68]. (**b**) Proximity ligation assay (PLA), used to study the interaction of the *p53* mRNA in the RING domain of HDM2 and the formation of the N-terminal HDMX-HDM2 heterodimer. This technique consists of two primary antibodies generated in different species that recognize the proteins of interest and two secondary antibodies that carry oligonucleotide sequences that are ligated and amplified, allowing detection by fluorophores [14]. (**c**) Proximity ligation ELISA (PLE) set to investigate the trimeric interaction among MDM2 and the nascent p53 peptide on isolated p53 polysomes (via RPL5 or RPL11). PLE involves three primary antibodies. One capture antibody (i.e., from goat), targeting one binding factor and a set of primary antibodies (from mouse and rabbit), targeting two additional binding factors. Upon capturing the trimeric complex via the capture antibody, recognizing one of the binding partners, signal amplification, and detection is performed following the principles of PLA.

**Figure 2 biomolecules-11-00036-f002:**
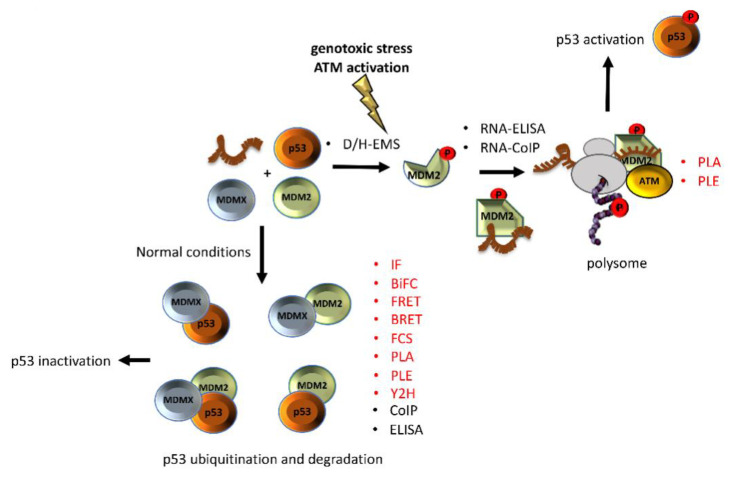
Different techniques have been used to unravel the interaction among of p53-MDM2-MDMX. A phosphorylation on Ser 395 of the MDM2 change its conformation and allow the binding of the *p53* mRNA that results in increased p53 protein synthesis after DNA damage. On the other hand, under normal conditions, interaction of MDMX and/or MDM2 with p53 negatively controls the p53 levels. Techniques that allow us to study the interactions in-situ are shown in red.

**Table 1 biomolecules-11-00036-t001:** Regulation of p53-MDM2 interaction studied with “classic” techniques.

Interaction	Condition	Technique	Results	PTM/Compounds	References
p53-MDM2	In vitro	EIA/ELISA	Importance of residues F19, W23, and L26 of p53 as contact points in the p53-MDM2 interaction	-	[30]
MDM2-p53	Oxidative stress in vitro	EI/ELISA	Reversible oxidation of Cys77 of Hdm2 disrupts inhibits the interaction with p53	NO	[31]
p53-MDM2 (S395D)	In vitro	ELISA, Co-IP	MDM2 S395D still binds p53 protein. The presence of p53mRNA block the interaction between the two proteins.	-	[14]
p53-MDM2	U2OS or H1299 cells treated with LMB	IF	MDM2-dependent ubiquitination and proteasomal degradation of p53 and identification of the subcellular localization where they take place.	-	[32,33]
ARF-MDM2	Ad-E2F1-infected NHF-1, U2OS, SJSA cells. Heterokaryion of HeLa and MEFs knock out (KO) for both p53 and MDM2 (2KO)	IF	Shuttling dynamics of the tumour suppressor ARF that explain its ability to stabilize p53.	-	[34,35]
MDM2-MDMX	MDM2 as a bait. In vitro translation. H1299 and JEG-3 cells.	Y2H, CoIP	p53 and MDM2 stabilization	-	[36]
MDM2-ribosomal protein S7	MDM2 as a bait. In vitro translation. COS7, U2OS, and A549 cells.	Y2H, CoIP	p53 stabilization	-	[37]
MDM2 oligomerization	SK-N-SH cells	BiFC (YFP)	RNA containing the IRES of *XIAP* inhibits MDM2 homodimerization and increases its stability	*XIAP* IRES	[38]
p53-MDM2	LNCaP cells γ irradiated	CoIP	Phosphorylation of p53 blocks interaction with MDM2		[29]
MDM2-RPS3 and p53-RPS3	H1299 and HEK293 cells with oxidative stress In vitro	CoIP, FRET, PLA	Induction of RPS3-MDM2-p53 complex formation under oxidative stress (H_2_O_2_) and RPS3-dependent inhibition of p53 ubiquitination	Nutlin-3	[39]

EIA/ELISA: enzyme immunoassay/enzyme-linked immunosorbent assay. IF: immunofluorescence. Y2H: yeast 2-hybrid. BiFC: Bimolecular Fluorescence Complementation. FRET: Förster Resonance Energy Transfer. BRET: Bioluminescence Resonance Energy Transfer. CoIP: Co-immunoprecipitation. ARF: Alternative Reading Frame p14 tumor suppressor. PLA: Proximity Ligation assay. MEF: Mouse Embryonic Fibroblast. RPS3: Ribosomal Protein S3.

**Table 2 biomolecules-11-00036-t002:** Examples of different molecules that disrupt or control the PPIs in the p53 pathway described using “classic” PPI techniques.

Interaction	Condition	Technique	Results	PTM/Compounds	References
MDM2-p53	Different constructs. In vitro translation.	Y2H, CoIP	Localization of a conserved region in the DNA-binding domain of p53 (Box II) that stabilizes the interaction with MDM2	Nutlin-3, MI-773, AMG232	[40]
MDM2-p53	Several constructs. U20S cells.	2H in mammals	Discovery of the p53-MDM2 inhibitor SL-01 that causes p53 stabilization and growth arrest in tumour cells	SL-01	[41]
MDM2-p53	PC-3 cells. Purified proteins. A549 cells.	BiFC. ELISA. CoIP.	Ceramide disrupts MDM2-p53 interaction in cells and in vitro	Nutlin-3, C_16_-ceramide	[42]
MDM2-MDMX	In vitro	HTRF-FRET	Inhibition of MDM2-MDMX interaction and E3 ligase activity toward p53	MMRi64	[43]
MDM2-p53 & MDMX-p53	In vitro. MCF7 cells.	TR-FRET. CoIP.	p53 activation through dual inhibition of MDM2 and MDMX	RO-5963	[44]
Hdm2-p53	H1299 cells	BRET	Validation of BRET to search compounds that disrupt PPIs using p53-HDM2 as a model	Nutlin-3	[45]

EIA/ELISA: enzyme immunoassay/enzyme-linked immunosorbent assay. Y2H: yeast 2-hybrid. 2H: 2 hybrid (in mammals). BiFC: Bimolecular Fluorescence Complementation. FRET: Förster Resonance Energy Transfer. BRET: Bioluminescence Resonance Energy Transfer. CoIP: Co-immunoprecipitation.

**Table 3 biomolecules-11-00036-t003:** p53 pathway interactions detected with new techniques.

Interaction	Condition	Technique	Results	PTM/Compounds	References
MDM2-p53	H1299 cells, doxorubicin	fluorescence cross-correlation spectroscopy	MDM2 binds p53 in an oligomerization-dependent fashion	Nutlin-3, MI733, RITA	[68]
RPS3-MDM2-p53	HEK 293 cells, oxidative stress	PLA, pull down, and FRET	RPS3 interacts with both p53 and MDM2. They suggest that RPS3 protect p53 from MDM2-dependent ubiquitination.	Nutlin-3	[39]
HDM2-HDMX-p53	H1299	PLA, Co-IP, and pull down	The triple complex between HDM2-HDMX-p53 involving the N-terminal region of the proteins increase p53 ubiquitination.	Nutlin-3a	[70]
HDM2-HDMX	H1299, DNA damage conditions with doxorubicin and etoposide	PLA, Co-IP, and pull down	After DNA damage, ATM phosphorylates both proteins inducing a change in conformation. This promotes the p53mRNA interaction in the RING domain of HDM2 and the N-termini HDMX-HDM2 heterodimer formation. The above results in HDMX and HDM2 increase degradation.	-	[14]

CoIP: Co-immunoprecipitation. PLA: Proximity Ligation Assay. FRET: Förster Resonance Energy Transfer.

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
