# Peer review of "Molecular and Biochemical Techniques for Deciphering p53-MDM2 Regulatory Mechanisms"

_biomolecules, 2020, doi:10.3390/biom11010036_

Round 1
Reviewer 1 Report
The manuscript entitled "Dissecting p53-Mdm2 interactome” by Karakostis et al., provided a short review on the research methods used to study protein-protein interactions (PPIs) in the p53-Mdm2 pathway. The authors summarized the classical techniques used to identify and characterize cellular PPIs, the fluorescent imaging methods used for studying endogenous protein interactions, and finally discussed the impact of post-translational modifications or ligand binding (proteins, RNA, or chemical compounds) on the interaction of p53 and Mdm2. This review presented some new technologies and recently developed PPI detection methods employed for understanding molecular interaction network using the p53-Mdm2 interaction as a model of cellular regulation. Although the manuscript is well written and presents an important and dynamic research topic, there are some specific concerns, which should be addressed to strengthen the manuscript before being accepted for publication.
- Because of its functional importance, p53 has been extensively studied, leading to identification of hundreds of p53 PPIs. These PPIs have been documented in literature and various databases (BioGRID database, Human Protein Reference Database, POINeT database). Due to the multifaceted and dynamic nature of p53 PPI network, an exhaustive list of p53-Mdm2 PPIs may not be possible. However, the present picture of p53-Mdm2 interactome described in this manuscript, is very limited and selective in scope. A more in-depth discussion of known p53-Mdm2 PPIs and better explanation of the PPIs focused in this manuscript should be included to reflect the current knowledge of this field.
- The term “interactome” implies a network of protein-protein interactions, and thus p53-Mdm2 interactome refers to the extensive network of interactions surrounding p53 and Mdm2. Recent advances in proteomics have enabled high-throughput measurement of cellular PPIs including the p53 interactome. However, the methods described in this manuscript focused mostly on the binary or selective protein-protein interactions in particular physiological context. Global approaches and large-scale proteomic strategies should be part of this review to provide a more comprehensive map of p53-Mdm2 interactome.
- Page 2, line 70-76. While the importance of translational control of p53 synthesis following genotoxic stress becomes better understood and more appreciated, description of the activation of p53 through phosphorylation and stabilization at the posttranslational level should be included in the manuscript.
- Some techniques are less common. Authors may consider to include more illustrations as Figure 1 to provide better visual depiction of different methods.
Author Response
Comments and Suggestions for Authors Report 1
The manuscript entitled "Dissecting p53-Mdm2 interactome” by Karakostis et al., provided a short review on the research methods used to study protein-protein interactions (PPIs) in the p53-Mdm2 pathway. The authors summarized the classical techniques used to identify and characterize cellular PPIs, the fluorescent imaging methods used for studying endogenous protein interactions, and finally discussed the impact of post-translational modifications or ligand binding (proteins, RNA, or chemical compounds) on the interaction of p53 and Mdm2. This review presented some new technologies and recently developed PPI detection methods employed for understanding molecular interaction network using the p53- Mdm2 interaction as a model of cellular regulation. Although the manuscript is well written and presents an important and dynamic research topic, there are some specific concerns, which should be addressed to strengthen the manuscript before being accepted for publication.
- Because of its functional importance, p53 has been extensively studied, leading to identification of hundreds of p53 PPIs. These PPIs have been documented in literature and various databases (BioGRID database, Human Protein Reference Database, POINeT database). Due to the multifaceted and dynamic nature of p53 PPI network, an exhaustive list of p53-Mdm2 PPIs may not be possible. However, the present picture of p53-Mdm2 interactome described in this manuscript, is very limited and selective in scope. A more in-depth discussion of known p53-Mdm2 PPIs and better explanation of the PPIs focused in this manuscript should be included to reflect the current knowledge of this field.
We thank the reviewer for his/her positive assessment of our work. We agree, of course, that the interactome of p53 is enormous and that it would not be possible to describe it all in one paper with specific scopes. As pointed by the reviewer, there are various databases that gather all the known interactions established by p53. To indicate this, we have added the following to the Introduction section (ll. 121-135):
“Diverse studies have employed proteomic techniques, aiming to identify such binding partners, aiding the unravelling of mechanisms regulated by the p53-MDM2 pathway[17-20]. However, in this review, we focus on well-studied mechanisms regulating the p53-MDM2 pathway, and on the experimental methodology that has been building up for the past years for revealing the mechanism whereby p53 is activated during the DNA Damage Response (DDR), rather list binding partners, which can be found in the literature and in databases, such as the BioGRID.”
Therefore, we focused on the regulation of the p53 pathway during the DNA damage response that is now described all along of the improved version of the manuscript.
The term “interactome” implies a network of protein-protein interactions, and thus p53- Mdm2 interactome refers to the extensive network of interactions surrounding p53 and Mdm2. Recent advances in proteomics have enabled high-throughput measurement of cellular PPIs including the p53 interactome. However, the methods described in this manuscript focused mostly on the binary or selective protein-protein interactions in particular physiological context. Global approaches and large-scale proteomic strategies should be part of this review to provide a more comprehensive map of p53-Mdm2 interactome.
We agree with the reviewer that the focus of the MS is about binary protein-protein interactions. This is because we deliberately put the attention on the mechanistic aspects that control the p53 pathway. Because of the lack of space, we are not able to elaborate on proteomics. However, as requested by the reviewer, we have included a short passage with some references to redirect the interested audience to very compelling and complete reviews and works (see above).
- Page 2, line 70-76. While the importance of translational control of p53 synthesis following genotoxic stress becomes better understood and more appreciated, description of the activation of p53 through phosphorylation and stabilization at the posttranslational level should be included in the manuscript.
As suggested, we have improved the MS and included a more explanatory description of the p53 activation at the posttranslational level. This is now better described in the Introduction (ll. 100-104), PPI classic techniques (ll. 157-159) and Proximity Ligation Assay (PLA) and the Proximity Ligation ELISA (PLE) sections in several parts as well as in the conclusion and perspective sections.
- Some techniques are less common. Authors may consider to include more illustrations as Figure 1 to provide better visual depiction of different methods.
We appreciate this observation. We now present an improved version of Figure 1 that includes the three most recent techniques.
Reviewer 2 Report
In this review, the authors have discussed the advantages and disadvantages of the techniques employed for the study of molecular interactions, focusing on the p53-MDM2 interaction. However, most of the part of the paper is devoted to the description of well-known techniques and well-known p53-MDM2 interaction, and therefore the amount of useful information in this paper is small. I hope the authors can divide each table into A. p53-MDM2 or MDMX interaction and B. other p53, MDM2 or MDMX interacting proteins, and explain more in detail about other p53, MDM2 or MDMX interacting proteins.
Reviewer 3 Report
The review work by Karakostis and colleagues strives to describe the available methodology to study protein-protein interactions. Due to the publication history, the authors focused, in large, on the methods that have been used to assess the p53-MDM2 interactions and the impact of posttranslational modifications.
As the authors specified, the major reason for doing so is that MDM2 and p53 belong to the family of proteins that contain intrinsically disordered regions (IDR).
The problem is that the methods described are the basic tools used to study protein-protein interactions. The referenced studies are rather generally described and it is difficult to conclude why the given method might be of a particular advantage while studying IDR.
All in all, this reviewer thinks that at the present form the work has low relevance to the filed.
Author Response
Comments and Suggestions for Authors Report 2
The review work by Karakostis and colleagues strives to describe the available methodology to study protein-protein interactions. Due to the publication history, the authors focused, in large, on the methods that have been used to assess the p53-MDM2 interactions and the impact of posttranslational modifications.
As the authors specified, the major reason for doing so is that MDM2 and p53 belong to the family of proteins that contain intrinsically disordered regions (IDR).
The problem is that the methods described are the basic tools used to study protein-protein interactions. The referenced studies are rather generally described and it is difficult to conclude why the given method might be of a particular advantage while studying IDR.
We thank the reviewer for raising this concern. As pointed by the reviewer, the described techniques can be used to study any PPI regardless of whether they contain IDRs. We have clarified this in the conclusion section by adding the following text (ll. 399-410):
“p53 is itself a very interesting molecule; it involves IDRs, dozens of PTMs sites and has tumour suppressor and transcription factor activities. These qualities make p53 a particularly challenging molecule, whose study requires the combination of several techniques. In the p53-MDM2 model, a single phosphorylation event in response to cellular conditions mediates a conformational change on the MDM2 protein with dramatic functional physiological effects on the regulation of p53. Such induced changes and their functional effects, may only be addressed by combining in cell, in vitro and in situ techniques, offering multifaceted insights (Figure 2). The techniques described here constitute technical cornerstones which drove the current knowledge around p53 regulation. However, these techniques are individually suitable for the study of any interaction, expression or mechanism independently on whether it involves IDRs, or PTMs inducing allosteric changes, in response to cellular stresses or exogenous factors. Clearly, each study may require a particular selection of the techniques to adequately address the regulatory mechanism.”
All in all, this reviewer thinks that at the present form the work has low relevance to the filed.
In the improved version of the MS, we have shifted the focus to the regulation of the p53 pathway, in particular during the DNA damage response. We now discuss how the described techniques have been used to study the mechanistic of the regulation, instead of simply portraying the techniques themselves. Specifically, the methodology described using the p53 pathway involves a set of techniques that constitutively can be employed to analyse any signalling pathway. It is a multifaceted methodology employing in vivo, in vitro, in cell and in situ techniques; and to our understanding it has a strong relevance to researchers and students working on p53 signalling, or regulatory mechanisms of various pathways. This methodology can be viewed as a valuable multifaceted technical toolset for analysing signalling pathways.

Reviewer 4 Report
The review “Dissecting the p53-MDM2 pathway” summarizes information on methods for studying the protein-protein interactions, as well as interactions of proteins with other biomolecules, using the example of two proteins, p53-MDM2. Provided information is interesting and very useful for specialists working in this field, as well as for a wider circle of researchers. The following comments can be made. The "Conclusion and Perspectives" section should be more clearly structured. It is possible to separate "Conclusion" and "Perspectives", highlight and separate aspects related to methods and data on specific interactions of two proteins - p53-MDM2 and so on. It is also necessary to add a list of abbreviations and not abuse their use in the text. It is better to replace the Running title “Dissecting the p53-MDM2 pathway” with a more accurate one that corresponds to the full title, for example, “The p53-MDM2 pathway”.
Round 2
Reviewer 1 Report
The revision is generally acceptable. However, the current manuscript still contains some minor issues as listed below.
- The authors have changed the title from “Deciphering the p53-MDM2 interactome” to “Deciphering the p53-MDM2 regulatory mechanism”. However, the content of this manuscript has nothing to do with the title. As this review focuses on the recently developed PPI techniques for understanding molecular interactions using the p53- Mdm2 interaction as a model of cellular regulation, the title should be modified to fit the theme, such as “ Development of molecular and biochemical techniques for deciphering the p53-MDM2 regulatory mechanism”.
- The abstract is difficult to follow and does not capture the focus and significance of this review. A more polished and organized piece of writing is expected in revision.
- Most information and discussion provided in conclusion should be placed in the introduction to strengthen the reasoning and logical order of the review. To make the writing more concise, the authors may consider to combine conclusions and perspectives.
Some minor points
Page 2/17 and line 55
“Their ability to interact with multiple partners is facilitated by intrinsically disordered regions (IDR), and it is often regulated by post-translational modifications (PTMs) that govern different conformations and bound states [7,8].”
As IDR is only one of the possible mechanisms, this sentence should be modified to the following.
“Their ability to interact with multiple partners is often facilitated by intrinsically disordered regions (IDR), and it is often regulated by post-translational modifications (PTMs) that govern different conformations and bound states [7,8].”
Page 2/17 line 74
“Activation of p53 requires MDM2 to switch and become a positive regulator of the p53 tumour suppressor protein. ”Should be
“Activation of p53 requires MDM2 to dissociate from and become a positive regulator of the p53 tumour suppressor protein”
Author Response
Reviewer 1
Comments and Suggestions for Authors
The revision is generally acceptable. However, the current manuscript still contains some minor issues as listed below.
Thank you! We are happy with this comment.
- The authors have changed the title from “Deciphering the p53-MDM2 interactome” to “Deciphering the p53-MDM2 regulatory mechanism”. However, the content of this manuscript has nothing to do with the title. As this review focuses on the recently developed PPI techniques for understanding molecular interactions using the p53- Mdm2 interaction as a model of cellular regulation, the title should be modified to fit the theme, such as “Development of molecular and biochemical techniques for deciphering the p53-MDM2 regulatory mechanism”.
We propose the next title based in the reviewer suggestion:
“Molecular and biochemical techniques for deciphering p53-MDM2 regulatory mechanism”
- The abstract is difficult to follow and does not capture the focus and significance of this review. A more polished and organized piece of writing is expected in revision.
We now present a polished version of the abstract.
- Most information and discussion provided in conclusion should be placed in the introduction to strengthen the reasoning and logical order of the review. To make the writing more concise, the authors may consider to combine conclusions and perspectives
We have improved the introduction to make it easy to follow. However, in the previous revision, we were asked by one reviewer to separate conclusions and perspectives.
Some minor points
Page 2/17 and line 55
“Their ability to interact with multiple partners is facilitated by intrinsically disordered regions (IDR), and it is often regulated by post-translational modifications (PTMs) that govern different conformations and bound states [7,8].”
As IDR is only one of the possible mechanisms, this sentence should be modified to the following.
“Their ability to interact with multiple partners is often facilitated by intrinsically disordered regions (IDR), and it is often regulated by post-translational modifications (PTMs) that govern different conformations and bound states [7,8].”
We changed the text accordingly
Page 2/17 line 74
“Activation of p53 requires MDM2 to switch and become a positive regulator of the p53 tumour suppressor protein. ” Should be
“Activation of p53 requires MDM2 to dissociate from and become a positive regulator of the p53 tumour suppressor protein”
We changed the text accordingly:
“Activation of p53 requires MDM2 to switch from binding the p53 protein to the p53 mRNA and become a positive regulator of the p53 tumour suppressor protein”.
Reviewer 2 Report
The authors’ claim to the reviewers’ comments are fairly acceptable, and the authors have revised the manuscript properly. I believe that this manuscript is now suitable enough to be published in Biomolecules.
Author Response
Reviewer 2
Comments and Suggestions for Authors
The authors’ claim to the reviewers’ comments are fairly acceptable, and the authors have revised the manuscript properly. I believe that this manuscript is now suitable enough to be published in Biomolecules.
We are very happy to know the reviewer is satisfied.
Reviewer 3 Report
The work has been improved. Authors have extended and change the angle of the manuscript so that it focuses on the methods to study PPIs with a particular focus on p53-MDM2 and p53-MDMX and not on intrinsically disordered proteins. Still, the authors often refer to the allosteric nature or shift in53 without providing a comprehensive description of the phenomenon.
Relating, the methods described are still the ones that the reviewer finds basic and not helpful in studying allostery in proteins.
There are also critical errors such as:
'CoIP can be coupled to 103 western blot to detect a specific interactor, to mass spectrometry in high-throughput settings and it
104 can also be linked to qRT-PCR to detect protein-nucleic acids interactions. However, CoIP performs
105 poorly for detecting low affinity liaisons. Since cells are disrupted, CoIP does not reveal the in situ 106 localisation of the interactions and has a significant risk of detecting false-positives due to artificial 107 or biologically non-relevant associations.'
The authors should extend this part to Chromatin immunoprecipitation, next, I cannot agree that low-affinity interactions cannot be detected with IP. It might require using customized conditions, however, the amount of the protein in the lysate can be modified in such a way that the detection is feasible.
In the end, BRET and FRET are methods established several years ago as well as FCS and FCCS. The authors, nonetheless, have not cited relevant literature eg for investigating the binding of the drugs to p53 using FCS or have not mentioned F2H assay used to study PPI.
Since many of the methods discussed therein have not been established by the authors' labs, this work lacks significance. It would have been far more interesting to the community if the authors strived to publish protocols/methods that they have developed/customized to study p53/MDM2/X interactions. Thus, the reviewer doubts the scientific value of this work and thus rejects the manuscript.
Author Response
Reviewer 3
Comments and Suggestions for Authors
The work has been improved. Authors have extended and change the angle of the manuscript so that it focuses on the methods to study PPIs with a particular focus on p53-MDM2 and p53-MDMX and not on intrinsically disordered proteins. Still, the authors often refer to the allosteric nature or shift in53 without providing a comprehensive description of the phenomenon.
Relating, the methods described are still the ones that the reviewer finds basic and not helpful in studying allostery in proteins.
Intrinsically Disordered Regions is a common feature among proteins, and the MDM2/MDMX–p53 axis constitutes a good example of how regulation of different IDRs via protein and RNA ligands as well as protein modifications control interactions and protein functions. There are few better studied examples. Due to overview nature of the article and limited space we have not gone in too much details but these can be found in the listed references.
We are not completely agreed with the reviewer since we know that certain methods can be used to study IPP and depends of the focus we can study Intrinsically Disordered Regions, or allosteric behaviour.
There are also critical errors such as:
'CoIP can be coupled to 103 western blot to detect a specific interactor, to mass spectrometry in high-throughput settings and it
104 can also be linked to qRT-PCR to detect protein-nucleic acids interactions. However, CoIP performs
105 poorly for detecting low affinity liaisons. Since cells are disrupted, CoIP does not reveal the in situ 106 localisation of the interactions and has a significant risk of detecting false-positives due to artificial 107 or biologically non-relevant associations.'
The authors should extend this part to Chromatin immunoprecipitation, next, I cannot agree that low-affinity interactions cannot be detected with IP. It might require using customized conditions, however, the amount of the protein in the lysate can be modified in such a way that the detection is feasible.
We now change the text accordingly, to include Chromatin immunoprecipitation as follow:
104 “can also be linked to qRT-PCR to detect protein-nucleic acids interactions in techniques such as CoIP-RNA and chromatin immunoprecipitation.”
CoIP is not suitable for studying weak interactions due to the washing procedures and changing the amount of total proteins will not change this. We change the text to take into consideration the point of view of the reviewer:
Page 3: “CoIP requires extensive washing steps with buffers including detergents to minimize unspecific interactions, and low affinity interactions might therefore be lost. The addition of crosslinkers can be used to stabilize weak interactions but at the same time runs the risk of causing unspecific interactions.”
In the end,
BRET and FRET are methods established several years ago as well as FCS and FCCS. The authors, nonetheless, have not cited relevant literature eg for investigating the binding of the drugs to p53 using FCS or have not mentioned F2H assay used to study PPI.
Thank you for this observation, we now include some cites and the F2H techniques, see pages 8-9 lines 244-254
Since many of the methods discussed therein have not been established by the authors' labs, this work lacks significance. It would have been far more interesting to the community if the authors strived to publish protocols/methods that they have developed/customized to study p53/MDM2/X interactions. Thus, the reviewer doubts the scientific value of this work and thus rejects the manuscript.
Detailed protocols can be found in the references. This is a review article and obviously we need to discuss methods developed by others! Why would the article lack significance because of this?